# Dual-Energy Computed Tomography-Based Quantitative Bone Marrow Imaging in Non-Hematooncological Subjects: Associations with Age, Gender and Other Variables

**DOI:** 10.3390/jcm11144094

**Published:** 2022-07-14

**Authors:** Florian Hagen, Jan Fritz, Antonia Mair, Marius Horger, Malte N. Bongers

**Affiliations:** 1Department of Diagnostic and Interventional Radiology, Eberhard-Karls-University, Hoppe-Seyler-Str.3, 72076 Tübingen, Germany; antonia.mair@yahoo.de (A.M.); marius.horger@med.uni-tuebingen.de (M.H.); malte.bongers@med.uni-tuebingen.de (M.N.B.); 2Grossman School of Medicine, NYU Langone Health, 550 First Avenue, New York, NY 10016, USA; jan.fritz@nyulangone.org

**Keywords:** bone marrow, age, gender, CRP, dual-energy computed tomography (DECT)

## Abstract

Background: Our aim is to assess the utility and associations of quantitative bone marrow attenuation (BMA) values measured on clinical dual-energy computed tomography (DECT) exams in non-hematooncologic subjects with skeletal regions, patient age, gender, and other clinical variables. Methods: Our local ethics committee approved this retrospective image data analysis. Between July 2019 and July 2021, 332 eligible patients (mean age, 64 ± 18 years; female, 135) were identified. Inclusion criteria were the availability of a standardized abdominopelvic DECT data set acquired on the same scanner with identical protocol. Eleven regions-of-interest were placed in the T11-L5 vertebral bodies, dorsal iliac crests, and femur necks. Patient age, gender, weight, clinical, habitual variables, inflammation markers, and anemia were documented in all cases. Results: Multi-regression analyses (all, *p* < 0.05) identified age as the strongest predictor of lumbar BMA (standardized coefficient: β = −0.74), followed by CRP (β = 0.11), LDH (β = 0.11), and gender (β = −0.10). In the lower thoracic spine, age was the strongest predictor (β = −0.58) of BMA, followed by gender (β = −0.09) and LDH (β = 0.12). In femoral bones, age was negatively predictive of BMA (β = −0.12), whereas LDH and anemia were positively predictive (β = 0.16 both). Heart insufficiency significantly decreased (β = 0.12, *p* = 0.034) a BMA value gradient from higher to lower HU values along the vertebrae T11 and L5, whereas age significantly increased this gradient (β = −0.2, *p* ≤ 0.001). Conclusions: DECT-based BMA measurements can be obtained from clinical CT exams. BMA values are negatively associated with patient age and influenced by gender, anemia, and inflammatory markers.

## 1. Introduction

Visualization and quantification of bone marrow have been a considerable limitation of computed tomography (CT) compared to magnetic resonance imaging (MRI), positron emission tomography (PET), and bone marrow scintigraphy [1,2,3,4]. However, dual-energy (DE) CT permits differential bone marrow evaluation based on three-material decomposition with calcium removal and bone marrow quantification (virtual non-calcium, VNCa), which can be used to detect and characterize focal and diffuse marrow lesions [5,6]. 

Physiological bone marrow composition may change throughout life and is further influenced by other factors, such as heavy smoking, obesity, physical activities, and anemia [7,8,9,10]. Histopathologic, experimental, nuclear medicine, and MRI-based techniques have differentiated longitudinal changes in bone marrow composition from hematopoietic marrow reconversion and bone marrow infiltration [11,12,13].

While MRI remains the modality of choice for imaging bone marrow and surrounding soft tissues due to its high contrast resolution [14], DECT is now employed in many patients with acute and chronic chest, abdomen, pelvis, and spine conditions [15]. Knowledge of DECT-based BMA patterns and associations with influencing factors may aid radiologists in more accurate and expedited diagnoses.

The purpose of our study was to assess the utility and associations of quantitative bone marrow attenuation (BMA) values measured on clinical dual-energy computed tomography (DECT) exams on virtual non-calcium images (VNCa) in non-hematooncologic subjects with skeletal regions, patient age, gender, and other clinical variables.

## 2. Materials and Methods

Our local institutional review board approved this retrospective data evaluation (registration number 246/2101BO2). The requirement for verbal and written informed consent was waived due to the retrospective nature of the study.

### 2.1. Patients

We performed a search of our radiological database for contrast-enhanced dual-energy CT exams of the abdomen and pelvis performed between July 2019 and July 2021 at our emergency department. 

Information obtained from our hospital information system included reasons for exam, demographics, past medical and surgical history, clinical diagnoses, follow-up, smoking history, alcohol abuse, diabetes mellitus, cardiac disease, and medical renal disease. 

Inclusion criteria were: (1) a standardized completed contrast-enhanced DECT exam, (2) no known rheumatological, hematooncological or metastatic tumor disease, (3) no vertebral body insufficiency fractures, and (4) no pronounced scoliosis. Exclusion criteria were: (1) a history of systemic anti-cancer treatment and (2) any hematooncological [16], metastatic tumor [17] or rheumatological [18] disease known to affect the bone or bone marrow (see Figure 1).

### 2.2. Computed Tomography Protocol and Post-Processing

All CT examinations were performed in supine patient position on a 3rd-generation SOMATOM Definition Force Dual-Energy 256-slice (Siemens Healthineers, Forchheim, Germany). The following examinational parameters were used: collimation of 0.6 mm, table speed 69.6 mm/s, table feed per rotation 23 mm, spiral pitch factor 0.6 matrix 512 × 512, convolution kernel QR40, 100 kVp and 150 kVp tube voltage, tube current time product 100 mAs (100 kVp) and 77 mAs (150 kVp), tin filtration for hardening of the high-energy spectrum. Owing to their higher frequency at our institution, we used contrast-enhanced DECT data, which allows differentiation between iodine and calcium, both in the phantom model and in clinical routine [19,20]. All exams were acquired using weight-adapted (1.2 mL/Kg/body weight) contrast medium volumes (Imeron 400 mg/mL (Bracco Imaging, Germany GmbH, Konstanz, Germany)) with a flow of 3 mL/s followed by a 40 mL saline chaser. The delay time was 65s.

Post-processing was performed using commercially available product software for dual-energy data, called “dual-energy bone marrow”, on syngo.via VB 40B (Siemens Healthineers). The post-processing software applied a three-material decomposition algorithm assuming that voxels within bone marrow could contain three material fractions with different X-ray absorption characteristics (fat, soft tissue, and mineralized bone), contributing to the total attenuation within the voxel. The total attenuation within the voxel could then be separated into a fat and soft-tissue partition and a calcium/bone mineral partition. In our case, calcium was removed from the calculation.

The software used the low (A series) and high-energy (B series) source data for input. Two output stacks of DICOM images were created, including mixed energy images containing information of both input series, resembling polychromatic single-source CT images with attenuation (HU) comparable to 120 kV tube energy, and a series of virtual non-calcium bone marrow images. Thresholds for the three-material decomposition were set as follows: soft tissue, 57/55 HU (low/high kV); fat, −103/87 HU; calcium slope, 1.44. 

### 2.3. Image Analysis

Two radiologists with two years (F.H.) and 30 years (M.H.) of experience in oncological imaging, respectively, first performed the image analyses independently. Mean CT values (HU) were calculated by averaging the CT values of the measurements for both readers. Substantial reader differences, such as positioning of regions of interest (ROI), were resolved by a final consensus session. Both radiologists were blinded to the medical history and the laboratory parameters.

First, the mixed dual-energy CT image (further referred to as standard CT image) data were evaluated for the presence of lytic, sclerotic, or mixed lytic-sclerotic metastatic bone disease.

Software-generated VNCa images were displayed as color-coded maps of the bone marrow. It was possible to adjust the degree of overlay between the color map and the CT image. A circular ROI tool was used to measure the mean absolute VNCa attenuation in eleven ROIs that were placed at the following sagittal sites in the vertebral bodies of the lower thoracic spine (T 11–12), upper lumbar spine (L 1–3) and lower lumbar spine (L 4–5), as well as in coronal plan for right and left dorsal iliac crests, and both femoral bones (see Figure 2A–C). 

The ROIs were created to be as large as possible, but at a minimum greater than 1 mm^2^. The ROIs were placed in the medullary bone while leaving a sufficiently wide margin towards the cortex and basi-vertebral veins due to frequent changes in the bone marrow consistency at these sites. The ROI tool delivered the mean attenuation in Hounsfield units (HU). 

We excluded vertebral bodies with height loss and lytic defects from the analysis. In six patients with hip arthroplasty implants, only the vertebral and iliacal crest BMA was included in the final analysis. 

### 2.4. Laboratory Data

We used our institutional normal reference values for red blood cells (RBC), 4.2–6.2 Mio/µL (female and male); hemoglobin, 12.0–16.0 g/dL (female), 14.0–18.0 g/dL (male); C-reactive protein (CRP) < 0.5 mg/dL (male and female); and lactate dehydrogenase (LDH) < 230 U/L. Anemia was defined as RBC < 4.2 Mio/µL and/or hemoglobin < 12.0/14.0 g/dL (female/male). 

### 2.5. Statistical Analysis

Statistical analysis was performed using SPSS (version 27.0.0, IBM, Armonk, NY, USA). Continuous data were expressed as means ± standard deviation (SD). Differences in patient characteristics between the two genders were assessed using an independent *t*-test or a Pearson Chi-Square test. For each patient, two scores were calculated: first, mean scores of the thoracic, lumbar, femoral and iliac bone marrow were calculated independently and subsequently used to determine the influence of the variables on an increased attenuation value of the bone marrow by a multivariate linear logistic regression model. In a second step, a weighted sum of the measurements of the seven vertebrae in the order from cranial to caudal (T11-L5) applying diametrical coefficients −3, −2, −1, 0, 1, 2, 3 resulting in a score of linear decline along the vertebrae. A positive score value indicated a decreasing BMA value gradient from T11 to L5, while a negative score indicated an increasing BMA value gradient from T11 to L5. Subsequently, this second score was used to determine the impact of the variables on the BMA ratio over the vertebrae by a multivariate linear logistic regression model. Linear logistic regression was calculated separately for the clinical laboratory markers and the clinical/habitual variables by age and gender. A one-way analysis of variance (ANOVA) was calculated to compare the attenuation values between the vertebral, iliac, and femoral bones. For statistical purposes and to establish standard values depending on age, patients were additionally sub-grouped in <30 years, >30 but <50, >50 but <70, and >70 years. Graphs were plotted on JMP (JMP 14, SAS Institute, Cary, NC, USA) and on Excel (Version 2206, Microsoft, Redmond, WA, USA). Statistical significance was set for *p* ≤ 0.05. 

## 3. Results

### 3.1. Subjects

The final study population consisted of 135 women and 197 men with a mean age of 64 ± 18 years. Further characteristics are displayed in Table 1.

### 3.2. Differences in Bone Marrow Attenuation by Age and Gender

Mean attenuation of the bone marrow differed significantly for every vertebral body between T11 and L5: −46.31 HU ± 27.77/−48.49 HU ± 26.50/−56.17 HU ± 27.66/−63.43 HU ± 27.49/−77.36 HU ± 29.85/−92.98 HU ± 29.72/−100.28 HU ± 29.58 (*p* < 0.001) (Figure 3).

All BMA values continued to a decrease over the years with an increasing gradient between the lower lumbar and the lower thoracic vertebral bodies (T11-L5) with increasing age (β = −0,2; *p* < 0.001) (Figure 4).

Gender had no significant prediction on this gradient over the years (*p* > 0.05).

Linear regression revealed the highest corrected R^2^ of 0.569 for the lumbar spine. Age was negatively associated with lumbar attenuation (standardized coefficient: β = −0.74, *p* < 0.001), followed by female sex (β = −0.10, *p* = 0.010). In the thoracic spine, age also was the highest negative prediction factor (β = −0.58, *p* < 0.001) followed by female sex (β = −0.09, *p* = 0.044), resulting in a corrected R^2^ of 0.358. In the femoral and iliac bone marrow, only increasing age had a significant negative association with BMA (β = −0.12, *p* = 0.027 and β = −0.27, *p* < 0.001). In terms of age subgroups, the highest difference was found between the 31–50 years and 51–70 years, irrespective of gender or skeletal region (see Table 2 and Figure 5).

### 3.3. Differences in Bone Marrow Attenuation by Inflammation Markers

CRP and LDH were the strongest positive predictors of lumbar spine BMA with β = 0.11 (*p* = 0.008) and β = 0.11 (*p* = 0.003). In the other bone marrow regions, only LDH had a significant positive association with BMA values indicated by β = 0.122 (thoracic spine), β = 0.145 (iliac crest) and 0.162 (femoral bone) (*p* < 0.05). Anemia had a significant positive association with BMA values of the femoral neck (*p* = 0.006, β = 0.156) (see Table 3).

Clinical laboratory markers of inflammation or anemia had association with the progressively increasing BMA gradient over time between the lower thoracic and lower lumbar vertebral bodies (*p* > 0.05).

### 3.4. Differences in Bone Marrow Attenuation by Other Clinical and Habitual Variables

The BMA values of the thoracic vertebral bodies had a significant negative association (*p* = 0.029) with heart failure (NYHA I-IV). None of the other clinical and habitual variables showed a BMA association in the thoracic spine or the iliac crest (see Table 4). Patients with diabetes type I showed a non-significant trend towards lower HU-values compared to patients without diabetes (*p* = 0.518). Mean patient age ranged between 51 years (diabetes type I) and 69 years (diabetes type II).

Moderately obese patients with a BMI > 35 kg/m^2^ (>class II) showed a non-significant BMA gradient from higher to lower values along the T11 and L5 vertebral bodies (β = −0.095, *p* = 0.084); however, this gradient was significant in patients with heart failure (NYHA I-IV) (β = 0.124, *p* = 0.034). The calculated linear model showed no effects in the other clinical variables on the age-dependent progressive pattern between T11 and L5 (*p* > 0.05).

## 4. Discussion

Our results show a negative correlation between BMA and age in the thoracolumbar spine and the femur bones, with small differences in the calculated significance levels. 

With increasing age, the hematopoietic (red) bone marrow becomes progressively replaced by fatty (yellow) marrow. The peripheral-to-central bone marrow conversion pattern in adulthood has been described for CT and MRI [21,22]. The physiological adult bone marrow distribution pattern may be present already at the age of 25, in which red marrow predominantly occupies the axial skeleton, sternum, and ribs [13,23]. However, bone marrow reconversion triggered by non-medical and medical conditions may occur at any adult age and alter this pattern [24]. Following the described physiological adult bone marrow pattern, we found a continuous, longitudinal decline in BMA in all analyzed age subgroups for both men and women. Notably, there were differences in BMA in the axial skeleton, with higher BMA values found in the lower thoracic spine compared to those measured in the lumbar spine and the pelvic bones. Interestingly, the calculated mean BMA values in the age subgroup 31–50 years were slightly higher in women, whereas this trend progressively inversed with increasing age, presumably due to hormonal changes. 

Irrespective of gender and analyzed bone marrow regions, the highest difference in BMA was found between the age subgroups 31–50 years and 51–70 years. These findings are the results from the work of Christy et al., who found that the percentage of active (hematopoietic) marrow in the body increased between birth and age 40 years in the thoracolumbar spine and pelvis, whereas in the femur and to a lesser extent in the humerus the percentage of red marrow continuously declined [25]. Differences in the distribution of red and yellow marrow throughout the axial skeleton, as reported by Schneider and Montz and Alfrey, confirmed these findings [26,27]. 

Notably, the mean BMA values in all age subgroups were negative, irrespective of the skeletal region. Accordingly, the mean values and ranges should guide routine CT-based bone marrow imaging in differentiating age-related bone marrow attenuation (composition) from hematopoietic marrow reconversion and, particularly, from bone marrow infiltration, e.g., by neoplastic tissue. 

Earlier studies have demonstrated that yellow marrow consists of approximately 95% fat cells and 5% non-fat cells, whereas hematopoietic marrow consists of 40% fat cells and 60% hematopoietic cells [28]. However, in certain conditions of increased hematopoietic demand, the reversal of physiological bone marrow conversion towards yellow marrow occurs. For this purpose, we also analyzed other potential factors that could influence these processes, including clinical and habitual variables and their impact on the BMA. Interestingly, smoking history and the presence of alcohol abuse, diabetes mellitus, and renal insufficiency did not significantly impact the mean BMA in our series. However, heart failure (NYHA I-IV) had a significant impact on the BMA of the thoracic vertebrae. The bone marrow–cardiac axis is an increasingly explored interaction, as the bone-marrow-derived progenitor cells participate in cardiac regeneration and functional recovery in the setting of progressive heart failure [29]. The known influence of heart failure on inflammatory markers on human bone marrow vascularization would lead to a bone marrow reactivation and thus higher BMA [30]. However, we found an opposite association between BMA and bone marrow, presumably because the negative age association predominated. Moreover, the reported conversion into yellow marrow in diabetic patients [31] could not be confirmed by our findings. The main reason might be the fact that changes of bone marrow fat in metabolic diseases were predominantly assessed in the specialty literature by single-voxel Magnet Resonance Spectroscopy for bone marrow fat quantification [31,32], which is by comparison more sensitive as CT. Furthermore, the advanced age and the associated increased bone marrow fatty degeneration appear to mask the influences of diabetes. Another reason might be the small size and heterogeneous sample of diabetic patients (*n* = 40), with diabetes type I as well as diabetes type II, treated by nutrition or by medication. However, a further subdivision would not be statistically meaningful due to the low strengths and should be addressed in another setting.

An increasing gradient between the BMA values measured in the lower thoracic spine vs. lumbar spine was noticed in obese patients and patients with known heart failure. However, these calculated values did not reach statistical significance. Some laboratory parameters, such as CRP and LDH, showed predictive values on the level of BMA in our cohort. CRP and LDH were the strongest predictors of lumbar spine BMA, whereas, in the other bone marrow regions, LDH demonstrated a significant association with BMA. This aspect was analyzed, as most patients in our cohort were referred for the CT exams due to acute symptoms suggesting infection or inflammation, both expected to have a certain reactive impact on the composition of bone marrow. Poulton et al. analyzed the MRI signal changes in the appendicular skeleton and described smoking- and obesity-related signal changes due to bone marrow reconversion in a subgroup of their analyzed cohort, using an accurate MRI technique [7]. However, the incidence of bone marrow reconversion depended on the intensity of the past smoking history.

Patients with anemia had significantly higher BMA values in the femoral neck regions. At these sites, the BMA values in adults are lowest as demonstrated, and therefore any minimal change becomes more obvious, contrary to the remainder of the axial skeleton, which still harbors hematopoietic (red) marrow. However, the increase in BMA caused by anemia did not reach statistical significance for the rest of the axial skeleton. 

Previous studies have addressed the heterogeneity of red and yellow marrow distribution [13]. Knowledge of differences in the distribution of BMA values along the axial skeleton should avoid error by considering, e.g., only one reference vertebra for indirect estimation of the mean bone marrow composition. Concerning the differences in the composition of bone marrow spaces in the axial skeleton versus the appendicular skeleton, Gurevitch et al. demonstrated differences in the mesenchymal cell count between the two, with relatively small numbers of mesenchymal cells in the tubular bones, which become exhausted rather early, losing their ability to provide a newly differentiated hematopoietic microenvironment and to maintain red marrow throughout the organism’s life [33]. Hence, only the cancellous bone is permanently occupied to a different extent by functioning hematopoietic tissue. Because of this knowledge, visualization of the marrow spaces within the spongy bone has gained attention, and its diagnosis was consequently directed to use of the MRI technique. 

Our study has some limitations. First, more subjects would be needed to establish generally applicable BMA means and ranges. However, this study was intended to assess associations of BMA values measured on clinical CT exams with other factors. Our results should be considered as preliminary, warranting larger studies. Second, the influence of clinical and, in particular, habitual variables, such as smoking and obesity, are difficult to assess due to the retrospective study design, which lacks detailed information about the duration of these risk factors. Third, a test-retest reproducibility was not assessed due to the study design. Forth, we used contrast-enhanced DECT data for BMA quantification of BMA. The circulating iodine contrast could have influenced the BMA values; however, the effect of iodine and calcium separation (virtual non-calcium, VNCa) based on DECT was shown to be feasible both in the phantom model and in clinical routine [19,20]. However, due to the short delay time and the expected poor vascular supply of fat-rich bone marrow, this effect was considered minimal and not relevant in this context and, in particular, when using this imaging technique.

## 5. Conclusions

In conclusions, DECT-based BMA measurements can be obtained from clinical CT exams. BMA values are negatively associated with patient age and influenced by gender, anemia, and inflammatory markers.

## Figures and Tables

**Figure 1 jcm-11-04094-f001:**
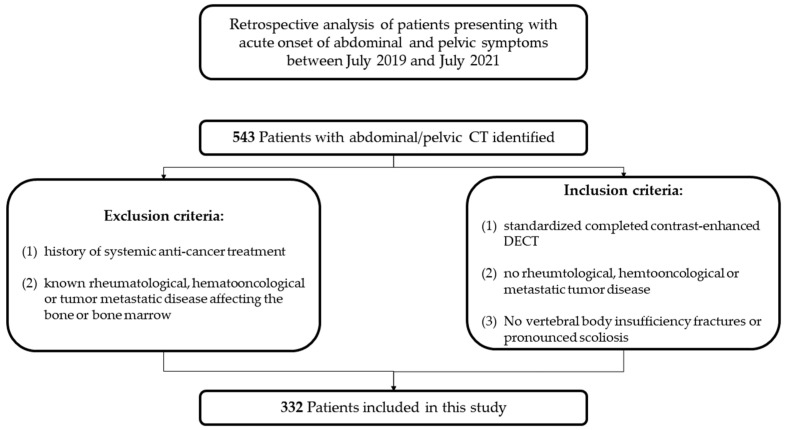
Study design flow diagram.

**Figure 2 jcm-11-04094-f002:**
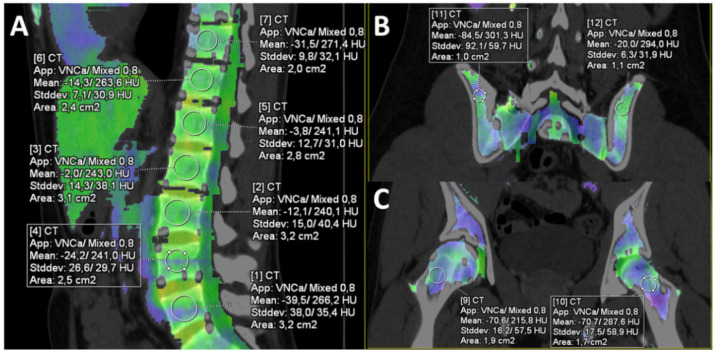
Region-of-interest measurements of bone marrow in a 20-year-old man presenting with acute appendicitis. (**A**) Region-of-interest measurements using a circle of at least 1 cm^2^ were placed in the T11 to L5 vertebral bodies. (**B**) Region-of-interest measurements in the iliac crests. (**C**) Region-of-interest measurements in the femoral necks. Yellow and green colored areas represent higher and blue and violet colored areas lower bone marrow attenuation.

**Figure 3 jcm-11-04094-f003:**
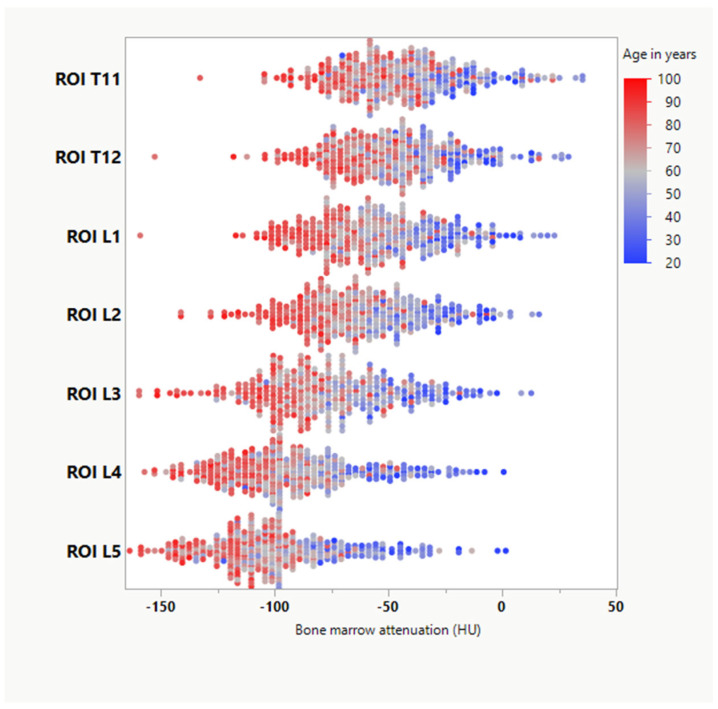
Scatter plot showing variations in bone marrow attenuation (BMA) depending on age and the vertebral body. ROI = region-of-interest. T11-L5 indicates the vertebral body of measurement.

**Figure 4 jcm-11-04094-f004:**
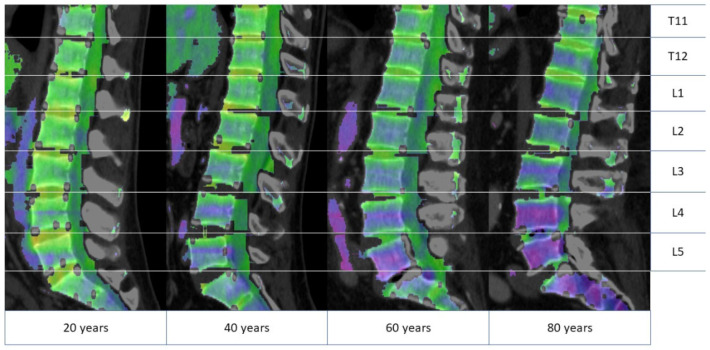
Bone marrow attenuation assessment of four different 20-, 40-, 60-, and 80-year-old male patients. The bone marrow attenuation is color-coded, with yellow and green colored areas representing higher and blue and violet colored areas lower bone marrow attenuation. The comparison demonstrates an overall age-associated decreasing bone marrow attenuation, which is disproportionately more pronounced in the lower spine. T11-L5 indicates the vertebral body level.

**Figure 5 jcm-11-04094-f005:**
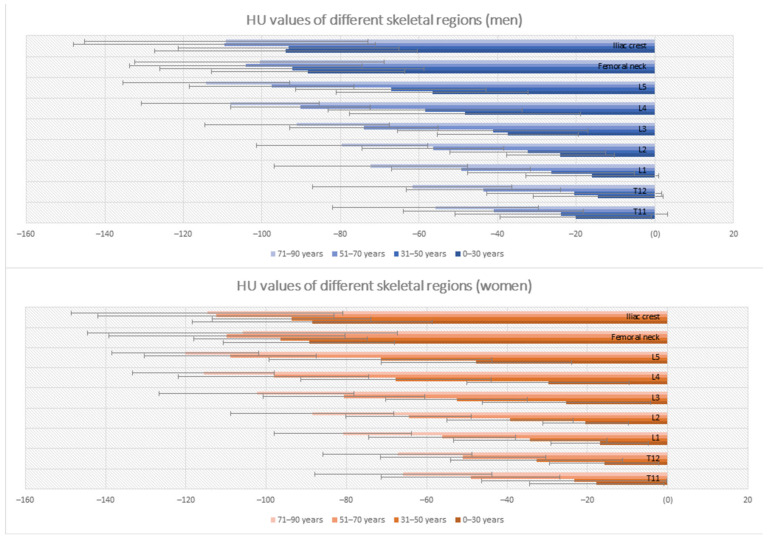
Bar graph showing variations in bone marrow attenuation (BMA) in HU depending on age groups and different skeletal regions. T11-L5 indicates the vertebral body of measurement.

**Table 1 jcm-11-04094-t001:** Patients’ characteristics.

	Male (197)	Female (135)	*p*-Value
Age in years (mean ± SD)	62 ± 17	67 ± 18	0.006 *
Weight in kg (mean ± SD)	81 ± 14	69 ± 14	<0.001 *
Size in cm (mean ± SD)	176 ± 7	165 ± 6	<0.001 *
$\mathrm{BMI}\;\mathrm{in}\;\frac{kg}{m^{2}}$(mean ± SD)	26 ± 4	25 ± 5	0.045 *
Heavy smoking history (n (%))	74 (37.6%)	24 (17.8%)	<0.001 **
Diabetes (n (%))	23 (18.8%)	17 (20.0%)	0.556 **
Heart failure (NYHA I-IV) (n (%))	21 (10.7%)	12 (8.9%)	0.596 **
Renal failure (n (%))	22 (11.2%)	11 (8.1%)	0.366 **
Alcohol abuse (n (%))	30 (15.2%)	4 (3.0%)	<0.001
$\mathrm{CRP}\;\mathrm{in}\;\frac{mg}{dL}$(mean ± SD)	6.8 ± 8.5	6.6 ± 9.7	0.814 *
$\mathrm{LDH}\;\mathrm{in}\;\frac{U}{L}$(mean ± SD)	260.6 ± 501.7	235.4 ± 166.7	0.575 *
Anemia (*n* (%))	68 (34.5%)	64 (47.4%)	0.018 **

* independent *t*-test; ** Pearson Chi-square test.

**Table 2 jcm-11-04094-t002:** Skeletal reference values (in HU) based on gender and age groups.

	0–30 Years		31–50 Years		51–70 Years		71–90 Years	
Vertebral Body	Men	Women	Men	Women	Men	Women	Men	Women
T11	−20.15 ± 19.27	−17.73 ± 16.73	−23.81 ± 27.07	−23.32 ± 23.06	−42.05 ± 22.93	−49.11 ± 22.28	−55.83 ± 26.25	−65.86 ± 22.08
T12	−14.44 ± 16.52	−15.85 ± 13.67	−20.53 ± 22.23	−32.74 ± 21.42	−43.61 ± 19.60	−50.99 ± 20.62	−61.72 ± 25.41	−67.30 ± 18.61
L1	−15.93 ± 16.88	−16.93 ± 12.10	−26.41 ± 21.26	−34.31 ± 19.13	−49.32 ± 17.65	−56.20 ± 18.36	−72.30 ± 24.59	−80.92 ± 17.18
L2	−24.00 ± 13.79	−20.45 ± 10.74	−32.41 ± 19.82	−39.27 ± 15.75	−56.42 ± 18.03	−64.55 ± 15.58	−79.63 ± 21.81	−88.59 ± 20.30
L3	−37.42 ± 17.91	−25.23 ± 20.92	−41.26 ± 24.18	−52.56 ± 17.64	−74.12 ± 18.87	−80.59 ± 20.10	−91.10 *± 23.44*	−102.43 ± 24.28
L4	−48.35 ± 29.41	−29.75 ± 20.21	−58.43 ± 24.69	−67.77 ± 23.72	−90.17 ± 17.77	−98.20 ± 23.70	−108.04 ± 22.65	−115.61 ± 17.62
L5	−56.62 ± 24.43	−47.70 ± 23.72	−67.13 ± 24.23	−71.54 ± 27.81	−97.51 ± 20.91	−108.97 ± 21.40	−114.92 ± 21.21	−120.16 ± 18.28
Femoral neck *	−88.27 ± 24.54	−89.36 ± 21.28	−92.37 ± 33.67	−96.52 ± 21.62	−104.18 ± 29.59	−109.80 ± 29.42	−100.62 ± 31.75	−105.92 ± 38.65
Iliac crest	−93.90 ± 33.42	−88.50 ± 29.89	−93.19 ± 28.10	−93,71 ± 19.18	−109.60 ± 38.45	−112.47 ± 29.40	−109.14 ± 36.03	−114.73 ± 33.86

* Missing values due to both side arthroplasty (post-traumatic): 3 males and 3 females.

**Table 3 jcm-11-04094-t003:** Bone marrow attenuation based on clinical laboratory markers of inflammation and anemia.

Variables	Thoracic Spine		Lumbar Spine		Femoral Bone		Iliac Crest	
	Standardized Beta Coefficient (β)	*p*-Value	Standardized Beta Coefficient (β)	*p*-Value	Standardized Beta Coefficient (β)	*p*-Value	Standardized Beta Coefficient (β)	*p*-Value
LDH	0.122	0.006	0.107	0.003	0.162	0.003	0.145	0.007
CRP	0.038	0.392	0.107	0.008	0.099	0.078	0.019	0.724
Anemia	0.015	0.739	0.042	0.257	0.156	0.006	0.083	0.128

LDH = lactate dehydrogenase, CRP = C-reactive protein.

**Table 4 jcm-11-04094-t004:** Mean bone marrow attenuation values based on other clinical variables.

Variables	Thoracic Spine		Lumbar Spine		Femoral Bone		Iliac Crest	
	Standardized Beta Coefficient (β)	*p*-Value	Standardized Beta Coefficient (β)	*p*-Value	Standardized Beta Coefficient (β)	*p*-Value	Standardized Beta Coefficient (β)	*p*-Value
Smoking History	0.010	0.833	−0.020	0.610	0.013	0.826	0.017	0.771
Alcohol abuse	−0.033	0.484	−0.046	0.231	0.057	0.330	−0.001	0.981
Diabetes	−0.057	0.211	−0.042	0.261	−0.034	0.555	−0.049	0.374
Heart failure (NYHA I-IV)	−0.103	0.029	0.017	0.661	−0.020	0.734	−0.016	0.981
Renal Insufficiency	0.059	0.217	0.002	0.963	-0.042	0.476	0.007	0.909

NYHA = New York Heart Association.

## Data Availability

Not applicable.

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
