# Peer review of "Dual-Energy Computed Tomography-Based Quantitative Bone Marrow Imaging in Non-Hematooncological Subjects: Associations with Age, Gender and Other Variables"

_jcm, 2022, doi:10.3390/jcm11144094_

Round 1
Reviewer 1 Report
Material and Methods:
1. more detailed presentation of the exclusion criteria: What diseases are suspected to influence bone density? Presumably all diseases with a metabolic component, including metabolic syndrome. However, these patients were included. What is the basis for the assumption that diseases have an influence or not ?
2. It is not clear to me why some vertebral bodies are ultimately weighted with a factor of +/- 1/2 or 3 in the summed calculation of the BMA.
Discussion: how good is the medication history? For example, for diabetic patients, there are many therapeutic regimens that differ significantly in terms of their medications. Wouldn't a much better division into subgroups be necessary in this regard?
Table 1: Significance levels
Author Response
- More detailed presentation of the exclusion criteria: What diseases are suspected to influence bone density? Presumably all diseases with a metabolic component, including metabolic syndrome. However, these patients were included. What is the basis for the assumption that diseases have an influence or not?
Thank you for your legitimate inquiry. Please excuse this vague formulation. We only included patients who presented acutely to the emergency department and were not known to have any rheumatological, hematooncological, or tumor metastatic disease. Neither did they have any kind of systemic treatment affecting hematopoiesis. These diseases and their associated therapies are known to have an impact on bone marrow [1–3]. We are aware of the fact, that the bone marrow represents a dynamic organ with continuous changes taking place with increasing age and altered hematopoietic needs. Experimental studies suggest that increased bone marrow (BM) activity is involved in the association between cardiovascular risk factors and inflammation in atherosclerosis [4]. The bone marrow is also a target of many pathological changes and diseases affecting its major components (e.g. adipocytes and red blood cells) [5]. However, the impact of diabetes is expected to play an inferior role at CT-imaging as this modality is less sensitive to subtle changes in bone marrow composition compared to MRI. All in all, diabetes is expected to increase the fat content of the bone marrow and thus inversely correlate with measured BMA. This could be shown by a non-significant trend towards lower density in patients suffering from diabetes type I compared to the non-diabetic population (-39.14 ± 16.95 vs. -48.09 ± 0.57) (see also response 3). The lack of tendency in type 2 diabetics is probably masked by the relatively high age and the associated increase in adipocytes. However, with regard to other studies, Sheu et al. reported a higher Bone Marrow Fat (BMF) in men with diabetes compared to non-diabetic men and stated that the correlation between BMF and Bone Mineral Density (BMD) differed by the diabetes status [6]. Moreover, studies on changes of bone marrow fat in metabolic diseases have predominantly used single‐voxel Magnet resonance spectroscopy for bone marrow fat quantification which is presumably more accurate as CT [7]. As there are multiple factors affecting the bone marrow fat concentration in subjects with metabolic syndrome (serum vitamin D concentration, antidiabetic medicine, waist circumference, serum ferritin, metabolic syndrome, imbalanced lipid metabolism, and abnormal liver function) [8] this specially subdivision should be address in another, more specific setting.
We added this point to our discussion (Page 11, line 285 – 295) and hope you might agree with it.
“Moreover, the reported conversion into yellow marrow in diabetic patients [6] couldn’t be confirmed by our findings. Main reason might be the fact that changes of bone marrow fat in metabolic diseases were predominantly assessed in the specialty literature by single‐voxel Magnet Resonance Spectroscopy for bone marrow fat quantification [6,7] which is by comparison more sensitive as CT. Furthermore, the advanced age and the associated increased bone marrow fatty degeneration appear to mask influences of diabetes. Another reason might be the small size and heterogeneous sample of diabetic patients (n=40) consisting of diabetes type I as well as diabetes type II treated by nutrition or by medication. However, a further subdivision would not be statistically meaningful due to the low strengths and should be address in another setting.”
References
- Lecouvet, F.E.; Larbi, A.; Pasoglou, V.; Omoumi, P.; Tombal, B.; Michoux, N.; Malghem, J.; Lhommel, R.; Vande Berg, B.C. MRI for response assessment in metastatic bone disease. Eur. Radiol. 2013, 23, 1986–1997, doi:10.1007/s00330-013-2792-3.
- Silva, J.R.; Hayashi, D.; Yonenaga, T.; Fukuda, K.; Genant, H.K.; Lin, C.; Rahmouni, A.; Guermazi, A. MRI of bone marrow abnormalities in hematological malignancies. Diagn. Interv. Radiol. 2013, 19, 393–399, doi:10.5152/dir.2013.067.
- Teke, H.Ü.; Cansu, D.Ü.; Korkmaz, C. Indications for bone marrow examinations in rheumatology. Rheumatol. Int. 2019, 39, 1221–1228, doi:10.1007/s00296-019-04312-w.
- Devesa, A.; Lobo-González, M.; Martínez-Milla, J.; Oliva, B.; García-Lunar, I.; Mastrangelo, A.; España, S.; Sanz, J.; Mendiguren, J.M.; Bueno, H.; et al. Bone marrow activation in response to metabolic syndrome and early atherosclerosis. Eur. Heart J. 2022, 43, 1809–1828, doi:10.1093/eurheartj/ehac102.
- Aaron, N.; Costa, S.; Rosen, C.J.; Qiang, L. The Implications of Bone Marrow Adipose Tissue on Inflammaging. Front. Endocrinol. (Lausanne) 2022, 13, 853765, doi:10.3389/fendo.2022.853765.
- Sheu, Y.; Amati, F.; Schwartz, A.V.; Danielson, M.E.; Li, X.; Boudreau, R.; Cauley, J.A. Vertebral bone marrow fat, bone mineral density and diabetes: The Osteoporotic fractures in Men (MrOS) study. Bone 2017, 97, 299–305, doi:10.1016/j.bone.2017.02.001.
- Karampinos, D.C.; Ruschke, S.; Dieckmeyer, M.; Diefenbach, M.; Franz, D.; Gersing, A.S.; Krug, R.; Baum, T. Quantitative MRI and spectroscopy of bone marrow. J. Magn. Reson. Imaging 2018, 47, 332–353, doi:10.1002/JMRI.25769.
- Ma, Q.; Cheng, X.; Hou, X.; Yang, Z.; Ma, D.; Wang, Z. Bone Marrow Fat Measured by a Chemical Shift-Encoded Sequence (IDEAL-IQ) in Patients With and Without Metabolic Syndrome. J. Magn. Reson. Imaging 2021, 54, 146–153, doi:10.1002/jmri.27548.
- It is not clear to me why some vertebral bodies are ultimately weighted with a factor of +/- 1/2 or 3 in the summed calculation of the BMA.
Thank you very much for your comment. Based on a long-year experience with CT-based bone marrow images in all kind of patients undergoing whole-body CT, where we observed some differences in bone marrow attenuation throughout the entire spine, we additionally addressed these potential gradients within the spine besides the absolute mean values. It was of interest for us, how the ratio of the vertebral body values varies throughout the different spine regions. For this purpose, our statistician advised us to perform a weighted sum with the help of coefficients [-3 -2 -1 0 1 2 3]. Following this assumption, we indeed found out that the attenuation values steadily decrease cranio-caudally.
We would like to briefly explain our statistical method using the following example.
Imagine you have values of 1,2,3,4,5,6,7 for the vertebral bodies. The mean value (M1 = 4) can change if all values increase by one step: 2,3,4,5,6,7,8 without the gradient changing. It would still be the case that the first one has the lowest value and then the values increase linearly (gradient unchanged), but the mean value (in the 2nd case) increases by 1 (M2 = 5).
It could also be that the values change from the initial situation (M1) to 8,7,6,5,4,3,2. Then the mean value increases (M2=4), but the gradient changes to the opposite (decreasing from 8 instead of increasing from 2).
- Discussion: how good is the medication history? For example, for diabetic patients, there are many therapeutic regimens that differ significantly in terms of their medications. Wouldn't a much better division into subgroups be necessary in this regard?
Thank you for this precious query. We included the medication history recorded at our emergency department. In the daily routine, all patients are asked about their medication. A total of 40 patients presented with diabetes type I or II (nutrition or medication therapy). Based on the small number, our statistician recommended not to divide the group further, otherwise the multivariate regression analysis would not have been feasible. Please find enclosed the subdivision of our patients suffering from diabetes:
|
|
No Diabetes [n=292] |
Diabetes Type I (insulin) [n=2] |
Diabetes Type II (Nutrition therapy) [n=11] |
Diabetes Type II (Medicinal therapy) [n=27] |
p-value |
|
Age [mean ± SD] |
63.25 ± 17.97 |
51.00 ± 14.14 |
69.55 ± 15.98 |
69.33 ± 12.43 |
0.159* |
|
Mean value spine (HU) [mean ± SD] |
-39.14 ± 16.95 |
-48.09 ± 0.57 |
-34.63 ± 14.65 |
-35.59 ± 20.65 |
0.518* |
|
Weighted sum spine (HU) [mean ± SD] |
-68.16 ± 25.18 |
51.07 ± 35.05 |
-74.01 ± 24.40 |
-80.15 ± 22.42 |
0.07* |
* ANOVA test
We hope you might agree with our response and the changes in our results (Page 10, Line 229 – 231) and discussion part (Page 11, Line 290 – 295)
“Patients with diabetes type I showed a non-significant trend towards lower HU-values compared to patients without diabetes (p=0.518). Mean patient age ranged between 51 years (diabetes type I) and 69 years (diabetes type II).”
“Furthermore, the advanced age and the associated increased bone marrow fatty degeneration appear to mask influences of diabetes. Another reason might be the small size and heterogeneous sample of diabetic patients (n=40) consisting of diabetes type I as well as diabetes type II treated by nutrition or by medication. However, a further subdivision would not be statistically meaningful due to the low strengths and should be address in another setting.”
- Table 1: Significance levels
Thank you for your suggestion. We added the requested p-values calculated with an independent t-test or the Pearson-Chi-Square test to differentiate between both genders. We hope you might agree with it.
Table 1. Patients characteristics.
|
|
Male (197) |
Female (135) |
p-value |
|
Age in years [mean ± SD] |
62 ± 17 |
67 ± 18 |
0.006* |
|
Weight in kg [mean ± SD] |
81 ± 14 |
69 ± 14 |
<0.001* |
|
Size in cm [mean ± SD] |
176 ± 7 |
165 ± 6 |
<0.001* |
|
BMI in [mean ± SD] |
26 ± 4 |
25 ± 5 |
0.045* |
|
Heavy smoking history [n (%)] |
74 (37.6%) |
24 (17.8%) |
<0.001** |
|
Diabetes [n (%)] |
23 (18.8%) |
17 (20.0%) |
0.556** |
|
Heart failure (NYHA I-IV) [n (%)] |
21 (10.7%) |
12 (8.9%) |
0.596** |
|
Renal failure [n (%)] |
22 (11.2%) |
11 (8.1%) |
0.366** |
|
Alcohol abuse [n (%)] |
30 (15.2%) |
4 (3.0%) |
<0.001 |
|
CRP in [mean ± SD] |
6.8 ± 8.5 |
6.6 ± 9.7 |
0.814* |
|
LDH in [mean ± SD] |
260.6 ± 501.7 |
235.4 ± 166.7 |
0.575* |
|
Anemia [n (%)] |
68 (34.5%) |
64 (47.4%) |
0.018** |
*independent t-test; **Pearson Chi-square test

Reviewer 2 Report
The authors sought to evaluate the usefulness and relationships between skeletal regions, patient age, gender, and other clinical variables and quantitative bone marrow attenuation (BMA) values measured on clinical dual-energy computed tomography (DECT) exams on virtual non-calcium images (VNCa) in non-hematooncologic subjects. The authors discuss a current problem in hematooncology. The findings showed that gender, anemia, and inflammatory markers all had an adverse impact on BMA readings, which were also significantly correlated with patient age. The subject is current and may be helpful in clinical practice by giving doctors a better opportunity to identify hematooncologic infiltration of the bone and bone marrow. The process is generally reliable from a scientific standpoint and repeatable. The presentation and discussion of the data and outcomes are excellent. When combined with the well-established MRI results, these data, in my opinion, may be helpful in discriminating between different disease conditions.
Author Response
The authors sought to evaluate the usefulness and relationships between skeletal regions, patient age, gender, and other clinical variables and quantitative bone marrow attenuation (BMA) values measured on clinical dual-energy computed tomography (DECT) exams on virtual non-calcium images (VNCa) in non-hematooncologic subjects. The authors discuss a current problem in hematooncology. The findings showed that gender, anemia, and inflammatory markers all had an adverse impact on BMA readings, which were also significantly correlated with patient age. The subject is current and may be helpful in clinical practice by giving doctors a better opportunity to identify hematooncologic infiltration of the bone and bone marrow. The process is generally reliable from a scientific standpoint and repeatable. The presentation and discussion of the data and outcomes are excellent. When combined with the well-established MRI results, these data, in my opinion, may be helpful in discriminating between different disease conditions.
Thank you very much for this positive feedback. We are planning another study in which we intend to correlate bone marrow attenuation at CT with signal intensity on MRI with regard to different diseases.